# Global biogeography of microbes driving ocean ecological status under climate change

Zhenyan Zhang [1,9], Qi Zhang[1,2,9], Bingfeng Chen[1], Yitian Yu[1], Tingzhang Wang[3], Nuohan Xu[1,2], Xiaoji Fan[3], Josep Penuelas [4,5], Zhengwei Fu[6], Ye Deng [7], Yong-Guan Zhu [7,8] & Haifeng Qian [1] ✉

Microbial communities play a crucial role in ocean ecology and global biogeochemical processes. However, understanding the intricate interactions among diversity, taxonomical composition, functional traits, and how these factors respond to climate change remains a significant challenge. Here, we propose seven distinct ecological statuses by systematically considering the diversity, structure, and biogeochemical potential of the ocean microbiome to delineate their biogeography. Anthropogenic climate change is expected to alter the ecological status of the surface ocean by influencing environmental conditions, particularly nutrient and oxygen contents. Our predictive model, which utilizes machine learning, indicates that the ecological status of approximately 32.44% of the surface ocean may undergo changes from the present to the end of this century, assuming no policy interventions. These changes mainly include poleward shifts in the main taxa, increases in photosynthetic carbon fixation and decreases in nutrient metabolism. However, this proportion can decrease significantly with effective control of greenhouse gas emissions. Our study underscores the urgent necessity for implementing policies to mitigate climate change, particularly from an ecological perspective.

Microbial population of the oceans, the largest environment on Earth, reaches billions per liter of seawater, with an overwhelming diversity and complexity that is governed by both abiotic and biotic factors[1]. Previous research has provided insights into the diversity, structure and dynamics of microbial communities, which exhibit clear vertical stratification, regional variability, and temporal fluctuations in the oceans[2–4]. Marine microbes play a critical role in adding, removing, and transforming organic and inorganic materials from seawater, and their complex interactions drive the global biogeochemical fluxes of major elements, including carbon, nitrogen and sulfur[5,6]. Functional traits, including the ability in biogeochemical cycling, have been widely recognized as important perspectives for research on the ocean microbiome[2]. Global surveys using high-throughput nucleic acid sequencing have provided comprehensive information about the

[1]College of Environment, Zhejiang University of Technology, Hangzhou 310032, PR China. [2]College of Chemistry & Chemical Engineering, Shaoxing University, Shaoxing 312000, PR China. [3]Key Laboratory of Microbial Technology and Bioinformatics of Zhejiang Province, Hangzhou 310012, PR China. [4]CSIC, Global Ecology Unit CREAF-CSIC-UAB, Bellaterra, 08193 Barcelona, Catalonia, Spain. [5]CREAF, Campus Universitat Autònoma de Barcelona, Cerdanyola del Vallès, 08193 Barcelona, Catalonia, Spain. [6]College of Biotechnology and Bioengineering, Zhejiang University of Technology, Hangzhou 310032, PR China. [7]State Key Laboratory of Urban and Regional Ecology, Research Center for Eco-environmental Sciences, Chinese Academy of Sciences, 100085 Beijing, PR China. [8]Key Laboratory of Urban Environment and Health, Institute of Urban Environment, Chinese Academy of Sciences, Xiamen 361021, PR China. [9]These authors contributed equally: Zhenyan Zhang, Qi Zhang ✉e-mail: hfqian@zjut.edu.cn

functional diversity, genomic potential and even transcriptomic activity of the microbial cycling of elements in the ocean[7,8]. In addition, the frequency of nutrient acquisition genes can be used as biomarkers of nutrient limitation in the ocean[9,10].

Compared to other ecosystems on Earth, oceans potentially have the most complicated and changeable environmental conditions on broad spatial and temporal scales; as such, they are highly vulnerable to changes in climate from intensive anthropogenic disturbances, such as emissions of greenhouse gases[11]. These changing environmental conditions can eventually influence the structure of the microbial community and organismal interactions[12,13]. Previous research has demonstrated the impacts of anthropogenic climate change on the marine microbial community, including the reorganization of the main taxa[12,14,15], loss of diversity[16,17], and modification of primary productivity[16–18]. With recent advances in our understanding of ocean biogeography mediated by climate change, several studies have also identified the variable responses of marine microorganisms to climate change from low- to high-latitude regions and the poleward shifts of some specific taxa[14].

Previous studies have raised questions regarding whether the intricate dynamics of microbial diversity, structure, and functional traits align spatially and how they collectively respond to climate changes over time. Although several earlier studies have explored these three ecological dimensions of microbial communities simultaneously[7,19,20], they did not offer a systematic indicator for assessing the ecological status of marine microbial communities. In this study, we gathered a substantial dataset of 953 ocean samples from an extensive metagenome sampling project, namely, Bio-GO-SHIP[21] (Fig. 1a and Supplementary Data 1), for taxonomic and functional annotations. For functional annotation, we compiled a database of biogeochemical marker genes associated with the core pathways of photosynthesis, carbon fixation, nitrogen metabolism, and sulfur metabolism (Supplementary Data 2). These marker genes are expressed by microbes and contribute to biogeochemical processes in the ocean[7]. By using the above dataset and database, we overviewed the variations in microbial communities in oceans and their links to environmental conditions. Subsequently, we constructed machine learning models for each microbial index and predicted their current distribution patterns in the global ocean. In this step, we also established ecological status, which serves as a composite representation of microbial communities, considering their functional traits, diversities, and structures. Finally, we harnessed machine learning techniques to elucidate how climate change might affect the future alteration of ecological status and to pinpoint the key drivers behind this transformation. This study is designed to bridge existing knowledge gaps in the biogeographic profiles of the global ocean under the influence of anthropogenic climate change. This topic holds significant importance in guiding management decisions and establishing effective policy objectives aimed at preserving the ecological integrity of our oceans. Furthermore, our comprehensive framework, encompassing ecological status, can be applied to other microbial ecosystems to assess and predict their resistance and resilience in the face of environmental or climate changes.

## Results

### Temporal and spatial variation in the ocean microbial communities

Bio-GO-SHIP[21] is an international multidisciplinary project in which metagenomic samples are collected under standard pipelines without size fractionation (a detailed sampling protocol can be found in the 'Methods' section). Here, we collected 953 metagenomic samples from 8 cruises in Bio-GO-SHIP across 26 Longhurst Provinces[22] that traveled from 2011 to 2020 (Fig. 1a and Supplementary Data 1). Notably, Longhurst Province was used to show the spatial variation in microbial profiles more clearly. Longhurst Province is a long-standing concept of

biogeochemical partitioning in the global ocean[23] and has also been used for spatial distribution patterns of chemical conditions[24], animals[25], protists[26] and so on. Our results showed that the richness, Shannon index and relative abundance of the top five phyla and biogeochemical marker genes clearly varied among the different samples across temporal and spatial scales (Fig. 1b and Supplementary Figs. 1–6). Specifically, Bacteroidetes had the highest coefficient of variation, while richness had the lowest one. Besides, coefficient of variation of genes involved in photosystem I, photosystem II, the Calvin cycle, and sulfur oxidation were greater than those of genes involved in other functions (Fig. 1c).

To determine the links between environmental conditions and the microbial community, we obtained ten environmental factors for each sample (Supplementary Data 3) from the Geophysical Fluid Dynamics Laboratory Earth System Model version 4 (GFDL-ESM4)[27]. These environmental factors include temperature, salinity, partial pressure of carbon dioxide ($pCO_2$), mixed layer depth, and concentrations of dissolved oxygen, nutrients (nitrate, phosphate, silicate, carbonate and iron); these factors have also been used in previous studies for evaluating the response of microbial communities to environmental change[7,16,18]. Consistent with microbial indices, these factors clearly varied among different samples (Supplementary Fig. 7) and contributed to temporal and spatial variations in both the structure and function of microbial communities (Fig. 1d, e).

### Mapping the microbial profiles in the global ocean

The basic logic for our work is as follows: 1) Environmental variables change on broad spatial (e.g., different ocean regions) and temporal scales, and this progress can undoubtedly be influenced by the combination of natural processes and anthropogenic activities in the ocean[28]. Notably, natural processes (e.g., density and currents) can also be influenced by anthropogenic activities. 2) Changes in environmental conditions can subsequently alter the microbial profiles of the ocean[1,5,8].

The tight links between environmental conditions and microbial communities have sparked questions regarding whether we can quantitatively predict microbial profiles in unknown oceans by using existing environmental data. To this end, we constructed a reliable regression model using machine learning for each microbial index (Supplementary Data 4). Our models simultaneously considered multiple environmental factors rather than a single stressor and were more reliable for predicting future conditions[16,18]. The results of hyperparameter tuning for the XGBoost random forest, linear regression, LASSO regression, and K nearest neighbor algorithms based on tenfold cross-validation showed that the random forest algorithm had the highest prediction performance (Supplementary Fig. 8 and Supplementary Data 5). Thus, we used the random forest algorithm with the best combination of hyperparameters to construct the final regression model for each microbial index (except richness because of the low $R^2$).

Based on the regression models, we quantitatively predicted and mapped microbial profiles in the global ocean (Supplementary Figs. 9–11) under the current (2023) environmental conditions, which were obtained from GFDL-ESM4. We observed that maps of microbial profiles clearly divided the global ocean into several parts, indicating the feasibility of defining a comprehensive ecological index based on microbial profiles.

### Definition of the ecological status of ocean microbial communities

A microbial community can be described by multiple ecological dimensions, such as diversity, structure, and functionality, as outlined above. These characteristics in turn determine the ecological status of communities in various habitats. In terrestrial ecosystems, Guerra et al. defined priority areas for soil nature conservation by synthetically

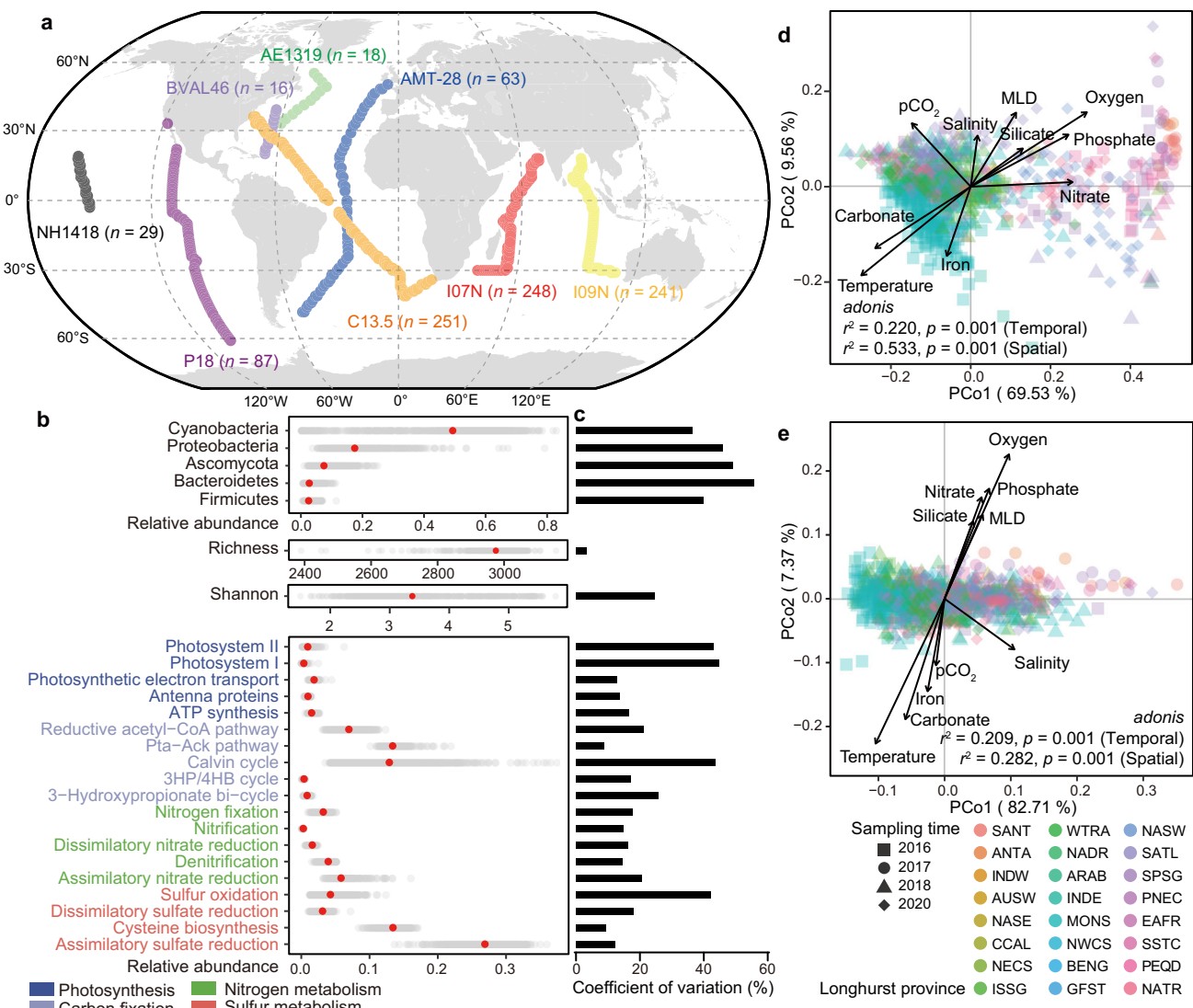

Fig. 1 | Environmental conditions contribute to the variation in microbial communities in the surface ocean. a Geographic distribution of samples in the Bio-GO-SHIP project. Each point indicates one sampling location to the nearest degree, with the point color indicating the metagenome sampling cruise. These cruises were performed at different times following different routes. b The structure, diversity, and functions of the microbial communities varied among the different samples. c Different microbial indices had different coefficients of variation. Specifically, Bacteroidetes had the highest coefficient of variation, while richness had the lowest coefficient of variation. The variation in genes related to photosystem I, photosystem II, the Calvin cycle, and sulfur oxidation was greater than

that in genes related to other functions. The structure (d) and function (e) of microbial communities exhibited clear spatial and temporal variation (evaluated by *adonis*, two-sided, $n = 890$), which resulted from changes in environmental conditions. The results of *adonis* are directly shown as $r^2$ and $p$ value in the plots. Spatial variation was evaluated by grouping samples into different Longhurst Provinces. A detailed description of each Longhurst Province can be found in Supplementary Data 1. MLD mixed-layer depth, ATP adenosine triphosphate, Pta-Ack phosphate acetyltransferase-acetate kinase, 3HP/4HB 3-hydroxypropionate/4-hydroxybutyrate.

considering soil biodiversity and ecosystem services[29]. Although some previous research has discussed the diversity, structure and functional traits of microbial communities simultaneously[7,19,20], we still lack a systematic indicator to describe the ecological status of microbial communities in the global ocean. This situation hampers our ability to understand the comprehensive influence of anthropogenic activities on microbial communities in ocean systems. In this study, we defined the ecological status of an ocean microbial community not only as representing taxonomic and diversity changes, as was done in previous studies[14,16], but also considering the biogeochemical potential (Supplementary Fig. 12).

Hyperparameter tuned hierarchical clustering successfully divided the ocean area (0.5° × 0.5° spatial resolution, $n = 44564$) into seven types of ecological status (defined as ES1 to ES7) (Supplementary

Data 6), which had significantly different microbial profiles from each other (Fig. 2a and Supplementary Data 7) and exhibited clear spatial variation (Fig. 2b). For example, most samples from ES4 existed in the tropical Atlantic Ocean and Indian Ocean. While ES7 dominated in the Antarctic area, ES2 was the main type of ecological status in the Arctic area. We also compared our results for 2023 with those for Longhurst Province[22]. The results showed that some Longhurst Provinces, such as the MONS and WTRA, had a single ecological status (Supplementary Fig. 13). However, the majority of Longhurst Provinces were comprised of multiple types of ecological statuses, indicating that the ecological status as defined in this study is a critical supplement for Longhurst Provinces.

The ecological status in this study could be used as a composite indicator to distinguish among ocean samples from a systematic

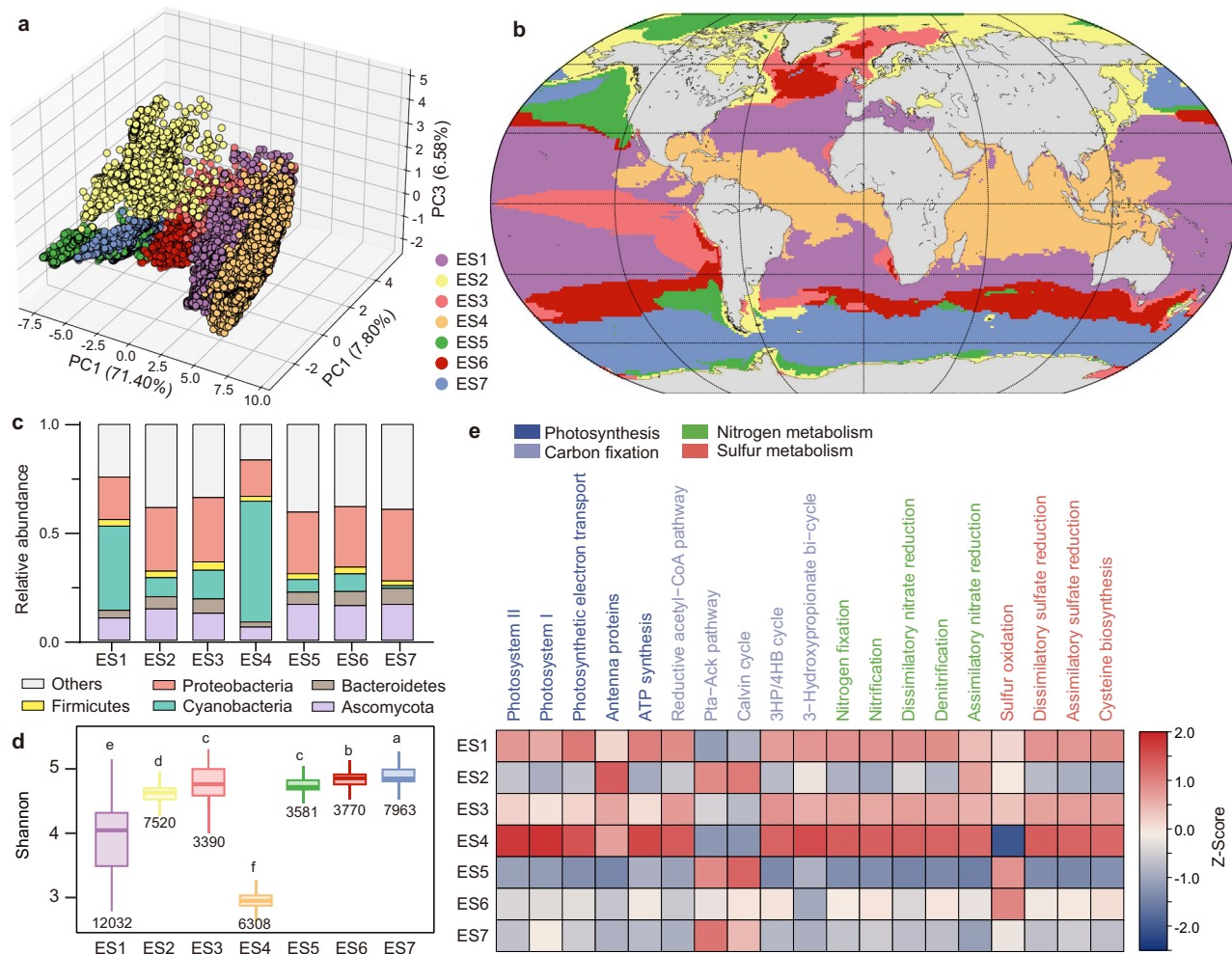

**Fig. 2 | Ecological status of microbial communities in the surface ocean.**
**a** Principal component analysis showing the distinct patterns of microbial communities in ocean samples belonging to different types of ecological status. The detailed pairwise comparisons for each microbial index between ecological statuses were performed by the Kruskal–Wallis $H$ test with Dunn's test, which can be found in Supplementary Data 7. **b** Global distribution of different types of ecological status under current environmental conditions. **c** Different types of ecological statuses exhibited different compositions of ocean microbial communities. Compared with the other ecological status types, ES4 had the highest abundance of Cyanobacteria. **d** Different types of ecological status had different microbial community diversities (represented by the Shannon index). ES7 had the highest diversity, while ES4 had the lowest. ES1 showed the median level of diversity. Data are presented as box plots with minima, lower quartile, center, upper quartile, maxima, and whiskers. The outliers of boxes were not shown. The number of biologically independent samples of each type of ecological status were shown under the boxes. Different letters represented the significant difference ($p < 0.05$) determined by Kruskal–Wallis $H$ test with pairwise comparisons (Dunn's test). **e** The biogeochemical potential of ocean microbes varied with different types of ecological status. In total, compared with the other samples, ES4 exhibited the highest functional potential in biogeochemical cycles; however, it had a low sulfur oxidation ability. Heatmap exhibited the relative abundance (scaled as Z-Score) of each pathway in each type of ecological status. ATP adenosine triphosphate, Pta-Ack phosphate acetyltransferase-acetate kinase, 3HP/4HB 3-hydroxypropionate/4-hydroxybutyrate.

perspective of the structure, diversity, and biogeochemical potential of the microbial community (Fig. 2c–e). Compared with the other ecological status groups, the ES4 group had the greatest abundance of Cyanobacteria, the lowest diversity, and the greatest functional potential for major biogeochemical processes. ES1 showed a median level of most microbial indices, including the median abundance for each dominant taxon, median diversity, and biogeochemical potential. Other types of ecological status all had low abundances of Cyanobacteria and were dominated by Proteobacteria; however, they can be distinguished from the perspective of diversity or biogeochemical potential. For example, ES3 and ES5 had no obvious difference in diversity (adjusted $p$ value > 0.05 in pairwise comparisons performed by Dunn's test, with |Cohen's d| < 0.1), but they were significantly different in terms of biogeochemical progress. Compared with ES3, ES5 had greater functional potential in the phosphate acetyltransferase-acetate kinase pathway (Pta-Ack pathway) and Calvin cycle, but its

functional potential in other pathways was much lower than that of ES3.

Overall, the ecological status clustered and defined here can be recognized as a composite description of ocean microbial communities, which we can easily use to construct a classification model and predict whole microbial alteration under changing environmental conditions. We applied machine learning classifiers using the same environmental factors in the regression model to predict the ecological status of the surface ocean (Supplementary Data 8). To reduce the bias resulting from different sample sizes for different types of ecological status, we performed random sampling for each type of ecological status ($n = 3000$). Compared to the original dataset, the randomly sampled dataset had the same composition of environmental factors (Supplementary Fig. 14) and correlations between them (Supplementary Fig. 15), and was then used as a training dataset in machine learning. And the remains were used as test dataset. The

results of hyperparameter tuning for random forest, XGBoost, support vector machine, and logistic regression based on tenfold cross-validation indicated that XGBoost, with $n\_estimators = 200$, $max\_depth = 9$, and $learning\ rate = 0.3$, had the highest prediction accuracy (Supplementary Data 9). Thus, we used the XGBoost algorithm with the above combination of hyperparameters to construct the final classified model (accuracy = 99.09% on the training dataset and 98.31% on the test dataset). The high prediction accuracy of the machine learning models we constructed also confirmed the reliability of the ten environmental factors in classifying ecological status. The confusion matrix also confirmed the good performance of the final model in the classification of ecological status (Supplementary Fig. 16). Besides, phosphate was the most important environmental factor in the prediction of the ecological status of the ocean (Supplementary Fig. 17).

### Changes in ecological status in the future ocean

Shared socioeconomic pathways (SSPs) induce a series of scenarios in which greenhouse gas (GHG) emissions (SSP1-1.9 (sustainability), SSP2-4.5 (middle of the road), SSP3-7.0 (regional rivalry), and SSP5-8.5 (fossil-fueled development without policy intervention))[30] contribute to the prediction of future changes in microbial communities across diverse habitats[14,16,29]. A range of distinct outcomes of climate change were observed for ten environmental factors at the end of this century compared to the present under different climate scenarios (Supplementary Figs. 18, 19). In the SSP5-8.5 scenario, which has the highest level of anthropogenic climate change with immoderate fossil fuel development and GHG emissions[30], the water temperature and $pCO_2$ increased, but the concentrations of phosphate, carbonate and oxygen decreased in the global ocean. In some specific areas, the concentrations of iron (tropical Pacific Ocean) and nitrate (tropical Atlantic Ocean and Indian Ocean) also increased under the SSP5-8.5 scenario. These changes in environmental conditions incontrovertibly altered the ecological status at the end of this century (Fig. 3a, b and Supplementary Fig. 20).

Our prediction indicated that the ecological status would change in 32.44% of the surface ocean area by the end of this century compared to the present level, because of changes in environmental conditions under the SSP5-8.5 scenario (Fig. 3b). However, if we positively implemented policies to mitigate climate change and control the full range of GHG emissions, such as those outlined in the Paris Climate Agreement[31], this proportion of changed areas in the surface ocean could effectively decrease (e.g., to 13.04% under the SSP1–1.9 scenario) (Fig. 3a). In addition, the proportions of different types of ecological status significantly changed in 2100 under the SSP5-8.5 scenario compared to the present scenario (Fig. 3c). Specifically, changes from ES1 to ES4 and from ES6 were common under the SSP5-8.5 scenario (Fig. 3d). However, environmental conditions under the SSP1-1.9 scenario only slightly altered the spatial distribution of each type of ecological status without obvious changes in their proportion in the global ocean (Fig. 3c and Supplementary Fig. 20), further confirming the necessity of implementing climate policies.

Using our regression models, we also focused on the detailed changes in diversity, key taxa, and biogeochemical potential between 2023 and 2100 under different climate change scenarios. We only considered ocean areas with changing ecological status (hereafter called "changed areas") in this analysis. Overall, the detailed microbial profiles exhibited results similar to those for the ecological status: the SSP5-8.5 scenario had greater impacts on all microbial indices (Supplementary Figs. 21–24). Changes in environmental conditions under the SSP5-8.5 scenario profoundly increased the abundance of Cyanobacteria in 61.70% of the changed area, mainly in the low-latitude regions (Fig. 3e and Supplementary Fig. 22). We also observed an increase in the functional potential for photosynthesis and carbon fixation (e.g., photosystem I, photosystem II, and the Calvin cycle) (Fig. 3f). These results were consistent with previous studies showing that elevated $CO_2$ and temperature under high GHG emissions

(Supplementary Fig. 19) promoted the growth of marine phytoplankton[32], primary production[33] and carbon fixation[34]. The results also revealed that the abundances of Proteobacteria, Ascomycota, Bacteroidetes and Firmicutes exhibited poleward shifts (increased in high-latitude regions but decreased in low-latitude regions) under future climate scenarios, especially under the SSP5-8.5 scenario (Supplementary Fig. 22), which has been widely mentioned in previous studies[12,14] as a thermal adaptation strategy for many microbes[17]. This poleward shift in the dominant taxa further resulted in considerable decreases in diversity (Supplementary Fig. 21), nitrogen metabolism (Supplementary Fig. 24c) and sulfur metabolism (Supplementary Fig. 24d) in low-latitude regions. However, poleward shifts in thermal adaptation cannot offset the negative impacts of future climate change on the biogeochemical potential of ocean microbes: more ocean areas will suffer from decreasing nitrogen and sulfur metabolism than from increasing nitrogen and sulfur metabolism (Fig. 3f).

### Environmental drivers in the alteration of ecological status

In the future, anthropogenic climate change may influence ecological status and microbial profiles by modifying environmental conditions. Consequently, the effects of anthropogenic climate change on changes in ecological status across each grid point of the surface ocean result from a combination of factors: i) the impacts of anthropogenic climate change on environmental factors and ii) the repercussions of these environmental factors on the alteration of ecological status. Although we have already identified phosphate as the most critical factor for predicting ecological status, these factors alone are insufficient for accurately evaluating the individual contributions of each factor to the alterations in ecological status induced by climate change across surface ocean grid points.

Previous studies have reported shifts in the dominant drivers of plankton biogeography reorganization under climate change across different oceanic regions[12,14]. In line with this, we also quantified the relative contributions of the ten environmental factors to the changes in ecological status induced by climate change, employing the methodology from these prior studies[12,14]. Anthropogenic climate change primarily reshaped the ecological status by altering nutrient concentrations (including nitrate, phosphate, silicate, iron and carbonate; 45.79% of the changed area) under the SSP1-1.9 scenario (Fig. 4a), while the dissolved oxygen and carbonate contents acted as two major drivers of the changing ecological status under the SSP5-8.5 scenario (Fig. 4b).

## Discussion

As the Earth's largest ecosystem, healthy oceans play a pivotal role in supporting human well-being through various means, such as food provision, livelihoods, recreational opportunities, and climate regulation[35]. Understanding the microbiome provides valuable insights into the role of the ocean in biogeochemical cycling during inevitable climate change. Developing an index for ocean health using microbial indicators is both essential and challenging[36] and requires comprehensive consideration from multiple perspectives. Recently, the concept of biogeochemical provinces has been widely used to partition the ocean by considering distinct patterns of biological variables and environmental conditions, such as primary productivity, chlorophyll-a concentration, taxonomic composition, temperature, and salinity[14,22,23]. This concept not only partitions the ocean but also enhances our understanding of the critical role of the ocean in global biogeochemical cycling. For example, compositional shifts among key planktonic groups in different oceanic provinces could be used to estimate the fluxes of nitrogen and carbon[14]. By using data from metagenome sampling projects, we systematically demonstrated the spatial and temporal variations in diversity, structure, and biogeochemical potential of ocean microbial communities and their response to changing environmental conditions. Most importantly, we propose

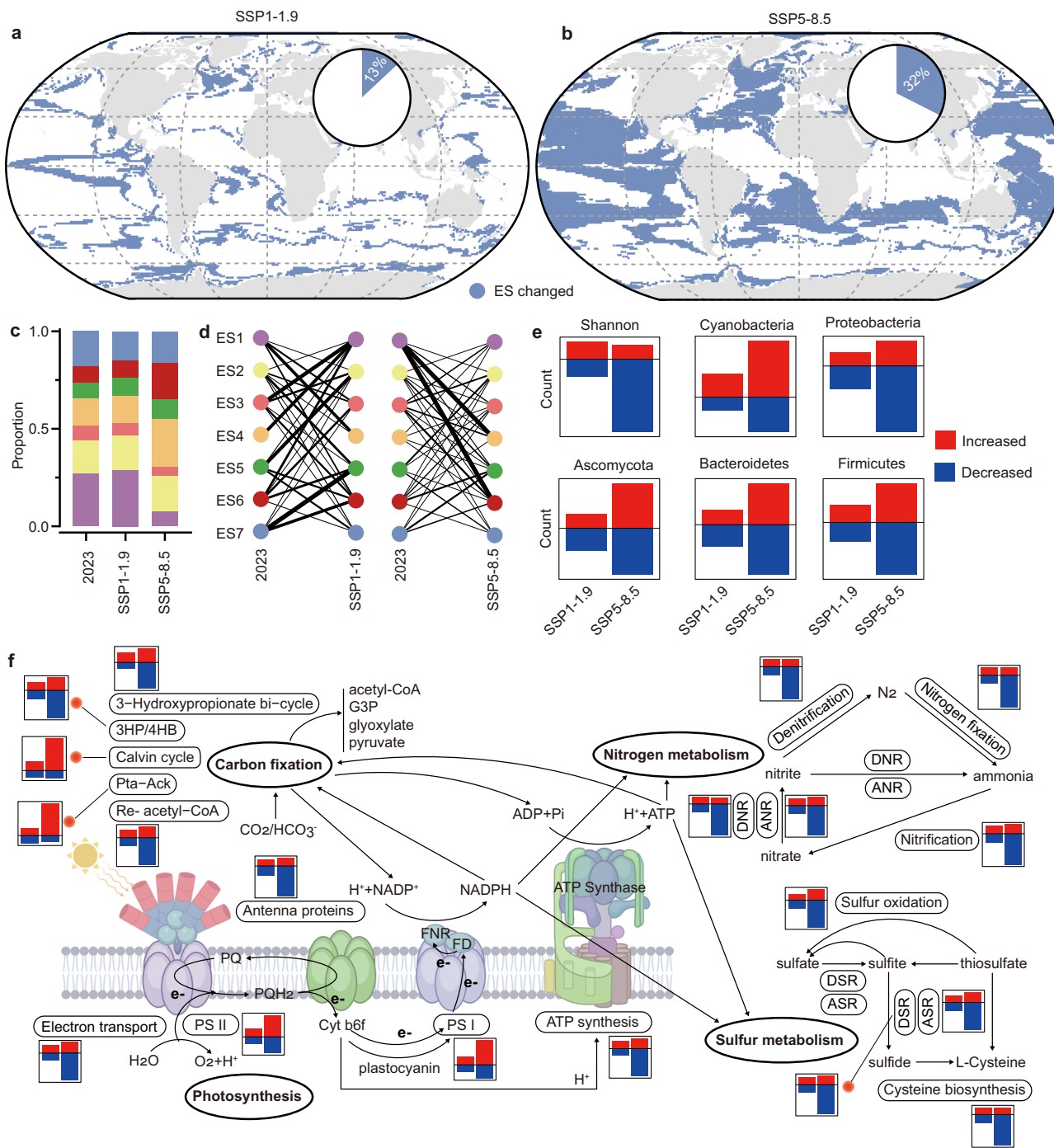

**Fig. 3 | Changes in the ecological status and detailed biogeographic profiles of ocean microbial communities in 2100 compared to those in 2023 under different climate change scenarios.** Changes in the ecological status of the global ocean in 2100 compared to that in 2023 under the SSP1-1.9 (**a**; 13.04%) and SSP5-8.5 (**b**; 32.44%) scenarios. Only the oceanic regions with changes in ecological status are shown here. The pie diagrams show the proportion of changed areas in the global ocean. The detailed changes in ecological status from 2023 to 2100 under the SSP1-1.9 and SSP5-8.5 scenarios can be found in Supplementary Fig. S20. **c** The proportion of each ecological status in the global ocean significantly changed in 2100 compared to that in 2023 under the SSP5-8.5 scenario. However, there were no obvious changes under the SSP1-1.9 scenario. **d** Different patterns of changes in ecological status under different climate change scenarios. Changes from ES1 to ES4 and ES6 were common under the SSP5-8.5 scenario. **e** Changes in the diversity and composition of ocean microbial communities under different climate change scenarios. **f** Changes in the biogeochemical potential of ocean microbes under different climate change scenarios. G3P glyceraldehyde 3-phosphate, Pta-Ack phosphate acetyltransferase-acetate kinase, 3HP/4HB 3-hydroxypropionate/4-hydroxybutyrate, Re- reductive, PS photosystem, ATP adenosine triphosphate, ADP adenosine diphosphate, NADPH and NADP+ reduced and oxidized nicotinamide adenine dinucleotide phosphate, FNR ferredoxin-NADP+ oxidoreductase, FD ferredoxin, PQH2 plastohydro quinone, PQ plastoquinone, Cyt b6f Cytochrome b6f complex, ANR assimilatory nitrate reduction, DNR dissimilatory nitrate reduction, ASR assimilatory sulfate reduction, DSR dissimilatory sulfate reduction.

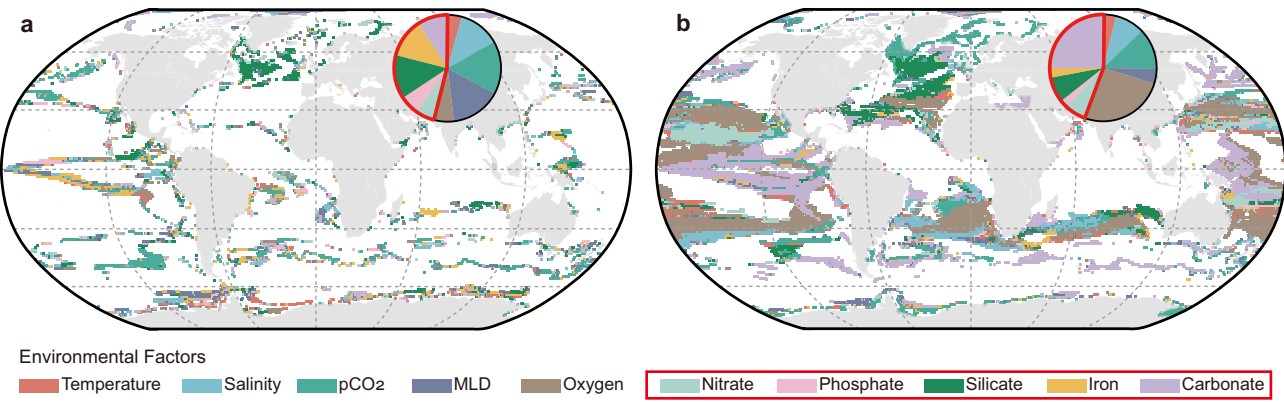

**Fig. 4 | Environmental factors contribute to climate change-induced changes in the status of the surface ocean.** The dominant environmental contributors to the changes in ecological status were determined at each grid point. The areas without changes in ecological status were excluded. The factors associated with nutrients are marked by a red frame and were the most important contributors under the SSP1-1.9 scenario (**a**). The ecological status in most ocean areas was determined by the concentrations of oxygen and carbonate under the SSP5-8.5 scenario (**b**). MLD mixed-layer depth.

a concept, ecological status, for ocean biogeography by integrating diversity, biogeochemical potential, and dominant microbial taxa.

Understanding how changes in environmental conditions, under the influence of climate change, impact microbial composition and biogeochemical progress in the ocean is paramount and enables us to take necessary actions to mitigate and adapt to climate change's effects on ecological systems. Based on machine learning, a sophisticated approach in current microbial research[14,16,29,37], we successfully predicted large-scale changes in the ecological status of the surface ocean from 2023 to 2100 under different climate change scenarios, confirming that the changing environmental conditions under anthropogenic climate change could eventually disrupt ecosystems[1,5,8]. While nutrient concentration was the most important driver of the changes in ecological status under the low-level climate change scenario, dissolved oxygen and carbonate contents altered the ecological status of the global ocean under high-level climate change scenario. Largely driven by the high level of anthropogenic activities (e.g., GHG emissions) under the SSP5-8.5 scenario, a decrease in dissolved oxygen in the global ocean is a widespread phenomenon and is predicted to continue throughout this century[38]. In addition, anthropogenic emissions of carbon dioxide also contribute to ocean acidification[39], which further causes a decrease in carbonate concentration[40]. Thus, as the concentrations of both oxygen and carbonate dramatically decreased in 2100 under the SSP5-8.5 scenario, ocean deoxygenation and acidification could play a critical role in the alteration of ecological status. Our results emphasized that positive climate policies are effective for mitigating alterations in the ecological status of the future ocean.

In addition, our predictions under different climate change scenarios also demonstrated that alterations in ecological status could be effectively summarized and represented by changes in detailed microbial profiles, including poleward shifts in the main taxa, increases in photosynthetic carbon fixation and decreases in nutrient metabolism. Thus, ecological status can be recognized as a more convenient and comprehensive index for evaluating the influence of anthropogenic activities on microbial communities in ocean systems. For better use of ecological status, we provide an easy-to-use tool called ES_predictor (available at https://doi.org/10.6084/m9.figshare.25627293), which can be used for determining the ecological status of ocean ecosystems with changing environmental variables under diverse anthropogenic pressures depending on the research goals. For example, if researchers have already measured the ten environmental variables in areas under different fishing pressures, they can easily evaluate the comprehensive impacts of fishing on microbial ecology

by our tools without metagenomic sequencing, which is costly and requires bioinformatic skills. Considering the amount of data currently available and in the future, this tool will also be continually refined.

However, it is essential to acknowledge that we cannot always perfectly predict and validate the effects of climate change on the ecological status of oceans by the end of this century. On the one hand, the metagenome samples in our dataset were collected from 61° S to 55° N (Supplementary Data 1), which may have resulted in a low prediction accuracy in ocean regions with absolute latitudes higher than 60°. On the other hand, the increasing occurrence of extreme events in global climatic systems[41,42] as well as extensive biotic events (e.g., Sargassum blooms in the Atlantic basin[43]) has induced complexity and uncertainty in future environmental conditions, which will play a decisive role in our prediction of ecological status. However, given the urgency of maintaining global ecological integrity and addressing climate change crises[44], we must operate under a degree of uncertainty[45]. This study provides up-to-date information to help scientists and policymakers collaboratively address these global crises. To improve our prediction of the ecological status of the ocean at high latitudes, future research needs to consider more sample collection from these regions under the standard pipelines referred to in Bio-GO-SHIP[21]. Additionally, global climate models, such as the GFDL-ESM4, should be properly amended based on the comprehensive understanding of the complicated link between climate change and extreme events[46]. Our predictions should also constitute a dynamic and long-term project under changing climate conditions resulting from updated global climate models. In addition to the above limitations, this study considered only the ecological status from a metagenomic perspective. We explored the functional potential of marker genes of core pathways involved in photosynthesis, carbon fixation, nitrogen metabolism, and sulfur metabolism. This approach proved useful and invaluable for overviewing and predicting the genetic potential of microbes in the biogeochemical processes of the global ocean and their response to climate change. However, different genes may affect biogeochemical processes to varying extents, and genetic potential is not entirely linked to transcriptomic activity, metabolomic composition, or element cycling. This limitation of our work could be addressed by employing multidisciplinary methods, such as transcriptomics, metabolomics, and chemometrics. However, at present, these approaches still lack international projects with standard sampling protocols.

## Methods
### Overview of ocean metagenome samples
We collected 953 metagenome samples of surface oceans from the large-scale metagenome sampling project Bio-GO-SHIP[21], which was

performed utilizing standard pipelines (Fig. 1a and Supplementary Data 1). The raw sequencing data were then downloaded from the NCBI-SRA database. Detailed information on the metagenomes is available in Supplementary Data 1, including the study ID/title, sample ID/title, sequencing strategy, laboratory, organization, isolation source, collection date, latitude, longitude, geological location and Longhurst Province code. Analysis with a bulk of metagenomic samples from public datasets is becoming increasingly available for making new biological discoveries; however, a series of biases caused by distinct protocols of sampling (e.g., size fractionation) and DNA extraction should be noted and reduced before analysis. In this study, we only used the dataset from Bio-GO-SHIP[21], which is a well-organized project and can minimize sample bias as follows (more details can be found in Larkin et al.[21]):

**Sampling protocol.** Taxonomic and functional traits are completely different for different fractionated microbes (for example, 0.22–3 μm for bacteria and 0–0.2 μm for viruses[14]), which may cause bias for samples with different size-fractionation schemes. In addition, samples from different depths cannot be compared if we only focus on the horizontal mapping of biogeographic patterns of ocean microbial communities (e.g., microbial profiles in different Longhurst provinces). For all the samples we used, researchers collected whole surface water (with a depth of ~3–7 m) into triple-rinsed containers and gently filtered it through a 0.22 μm pore size Sterivex filter (Millipore, Darmstadt, Germany). After filtration, the DNA was preserved with lysis buffer (4 mM NaCl, 750 μM sucrose, 50 mM Tris-HCl, 20 mM EDTA) and stored at −20 °C before extraction.

**DNA extraction protocol.** A previous study demonstrated that the method of DNA extraction had a considerable effect on the outcome of metagenomic analysis[47]. Fortunately, DNA extraction from each sample via Bio-GO-SHIP[21] was performed via the same method. Sterivex filters were incubated with lysozyme and proteinase K and SDS buffer. DNA was then extracted from the Sterivex filters using TE buffer (10 mM Tris-HCl, 1 mM EDTA), precipitated in an ice-cold solution of isopropanol and sodium acetate, centrifuged and resuspended in TE buffer for 30 min. Afterwards, the DNA was purified using a genomic DNA Clean and Concentrator kit (Zymo Research Corp., Irvine, CA). Finally, DNA concentrations were quantified using a Qubit dsDNA HS Assay kit and a Qubit fluorometer (Thermo Fisher, Waltham, MA).

### Annotation and calculation of the abundance of marker genes and taxa
For functional annotation, we compiled a database of biogeochemical marker genes associated with the core pathways of photosynthesis, carbon fixation, nitrogen metabolism, and sulfur metabolism (Supplementary Data 2). These marker genes can also be expressed by microbes and contribute to biogeochemical processes in the ocean[7]. We generated the nucleotide sequences for each marker gene from the Kyoto Encyclopedia of Genes and Genomes database (KEGG; release 109.0) and clustered them with 100% sequence similarity using CD-HIT[48] (v4.7) to construct a database of biogeochemical marker genes.

We used FastQC (v0.11.5; https://github.com/s-andrews/FastQC) to qualify the raw data of the metagenomic samples and Trimmomatic[49] (v0.36) to trim and filter the low-quality reads and obtain clean data for further analysis. The reads were annotated according to the biogeochemical marker genes in the database that we compiled using BWA[50] (v0.7.13), a fast and accurate short-read alignment tool. We removed unmapped reads using Samtools (v1.3.1)[51], a flexible tool for handling sequence alignment/map format, and then counted the number of mapped reads of cycling genes in each ocean sample using a script available on GitHub (https://github.com/ZhenyanZhang/Ecological_status)[52]. Additionally, we estimated the average genome size (AGS) by MicrobeCensus (v1.1.1)[53] based on the

abundance of 30 essential single-copy genes and calculated the genome equivalent for each metagenome sample as follows:

$$\text{Genome equivalent} = \frac{\text{Library size (bp)}}{\text{AGS (bp)}} \quad (1)$$

where library size is the total number of sequences. Finally, the abundance of each cycling gene was normalized as reads per kilobase per genome equivalent (RPKG):

$$\text{RPKG} = \frac{\text{Mapped reads}}{\text{Gene length (kbp)} \times \text{Genome equivalent}} \quad (2)$$

The use of RPKG can improve the detection of differentially abundant genes between metagenome samples[53]. Taxonomic annotation was performed by Kraken2[54] (v2.1.2) with the default parameters, and the abundance of each taxon was also normalized by RPKG.

### Links between microbial profiles and environmental factors
The coefficient of variation of each microbial index was defined as the standard deviation divided by the mean. The Longhurst Province, as a long-standing concept of biogeochemical partitioning in the global ocean[22], was used to show the spatial distribution patterns of microbial communities more clearly in this study. We obtained the Longhurst province information for each sample by using ArcGIS (v10.8). The sampling time and site from GFDL-ESM4 were searched to obtain the environmental factors for each sample. GFDL-ESM4 from the National Oceanic and Atmospheric Administration (NOAA) provided the present and future values of all ten environmental factors under different scenarios and was also used in a previous study for predicting the reorganization of plankton biogeography under climate change[14]. These environmental factors include temperature, salinity, partial pressure of carbon dioxide (pCO$_2$), mixed layer depth, and concentrations of dissolved oxygen, nutrients (nitrate, phosphate, silicate, iron) and carbonate, which have also been used in previous studies for evaluating the response of microbial communities to environmental change[7,16,18]. In total, we collected all ten environmental factors from 890 samples, and the other 63 samples collected before 2015 were not considered in further analysis. The Bray–Curtis distances of the taxonomic and functional traits of the microbial community in the ocean samples were calculated based on the abundances of the genera and genes, respectively, using the 'vegan' R package[55] (v2.5–7). To link the temporal and spatial variation of microbial communities with environmental factors, principal coordinate analysis with Bray–Curtis dissimilarity of the taxonomic and functional traits of the microbial community and its links to environmental factors were calculated using the 'vegan' R package[55] (v2.5–7) and visualized by the 'ggplot2' R package[56] (v3.4.2).

### Prediction of each microbial index in the global ocean
A series of regression models were used to predict the microbial indices in the ocean combined with tenfold cross-validation for better performance and less overfitting[37]. The ten environmental factors mentioned above were used as independent variables for machine learning in this study, while 26 microbial indices (5 dominant taxa, richness, Shannon indices and 19 biogeochemical pathway indices) were included as dependent variables for machine learning. In other words, we presumed that changing environmental conditions directly determine the microbial communities rather than the time and site of the samples. This premise ensures that we can predict microbial profiles and ecological status by using changing environmental data.

To identify the most suitable algorithm and related hyperparameters for machine learning, we first constructed regression models

using different algorithms (XGBoost random forest, linear regression, LASSO regression, and K nearest neighbors) with different combinations of hyperparameters with tenfold cross-validation. In hyperparameter tuning for each algorithm, we used only 80% of the samples as the training dataset, and the remaining data were used as the test dataset. After hyperparameter tuning, the best hyperparameter combination for each algorithm and each microbial index was further evaluated by the prediction performance (represented as $R^2$) on the overall dataset and test dataset. In total, random forest had the highest prediction performance for most of the microbial indices and was used to construct the final models ($R^2$ on the test dataset ranged from 0.198 to 0.844), which were saved for further prediction. However, we failed to construct a promising regression model for richness with all combinations of the hyperparameters and algorithms ($R^2$ on the test dataset ranged from −0.026 to 0.025). Thus, we did not consider richness in the following analysis and only used the Shannon index ($R^2$ on the test dataset = 0.809) to represent diversity. After construction of the final models, we also determined the relative importance of each environmental factor in the prediction of each microbial index. All machine learning steps, including hyperparameter tuning, algorithm selection, model construction and feature importance determination, were performed with the 'sklearn' Python package (v1.1.3)[57].

To delineate the microbial profiles in the current ocean, we obtained ten environmental factors in 2023 (average value in monthly data) from GFDL-ESM4 and predicted each microbial index with a 0.5° × 0.5° spatial resolution and one-year temporal resolution using the final models in the 'sklearn' Python package (v1.1.3). Maps of each microbial index in the global ocean were generated by the 'Basemap' Python package (v1.3.6).

### Determining the ecological status of the ocean samples

To further determine how the diversity, structure, and biogeochemical potential of the microbial communities respond to climate change, we defined the ecological status as a composite description of the ocean microbial communities. We constructed a matrix that included 1) the abundance of the five dominant taxa in the ocean, which constitute >80% of the microbial community; 2) the Shannon index of the ocean microbial community, which is a critical index of diversity; and 3) the abundance of each pathway, which represents the biogeochemical potential of the microbial community. Before performing hierarchical clustering, a principal component analysis was conducted with the 'sklearn' Python package to reduce the dimensionality of the data and increase the clustering performance. Only the principal components that account for 95% of the variance are retained for clustering. Hierarchical clustering was used to cluster samples with hyperparameter tuning (including different cluster numbers, clustering algorithms and distance metrics) in the 'sklearn' Python package (v1.1.3). The silhouette score, an important internal evaluation index in hierarchical clustering, was used to evaluate clustering performance. Finally, the 'ward' clustering algorithm with Euclidean distance was chosen to generate seven clusters, which were defined as ecological status in this study. We then analyzed the detailed global distribution and microbial profiles for each ecological status.

### Construction of a machine learning classifier for ecological status

To predict the ecological status in the ocean under future climate change scenarios, we also constructed a machine learning classifier for ecological status with ten environmental factors. The ten environmental factors above were used as independent variables for machine learning in this study, and the seven ecological statuses were included as dependent variables for the machine learning classifier. To reduce the bias resulting from different sample sizes for different types of ecological status, we performed random sampling ($n$ = 3000) for each

type of ecological status. Principal component analysis and correlation metrics between features were used to ensure the validity of the randomly sampled dataset (Supplementary Figs. 14, 15), which was then used as a training dataset in machine learning. The remaining dataset after random sampling was used as an independent test dataset.

To identify the most suitable algorithm and related hyperparameters for machine learning, we first constructed a classification model using different algorithms (random forest, logistic regression, support vector machine, and XGBoost) with different combinations of hyperparameters. Tenfold cross-validation was used for machine learning to ensure the performance of the random forest algorithm and avoid overfitting[37]. XGBoost with *n_estimators* = 200, *max_depth* = 9, and a *learning rate* = 0.3 was then chosen because it was more accurate (average accuracy = 99.09% in tenfold cross-validation) than the other algorithms and hyperparameter combinations (average accuracy ranging from 60.64% to 98.28%). The final classification model was then built using the 'sklearn' Python package (v1.1.3). The confusion matrix was also used to confirm the good performance of the final model in the classification of ecological status.

### Machine learning prediction

For predicting changes in ecological status and microbial profiles, we collected ten environmental factors used for machine learning with 0.5° × 0.5° spatial resolution and one-year temporal resolution for the surface ocean from GFDL-ESM4[27] under the SSP1-1.9 and SSP5-8.5 scenarios of climate change with different levels of trajectory of greenhouse gas (GHG) concentrations. We then predicted the changes in ecological status and microbial profiles in the surface ocean with the final models using the 'sklearn' Python package (v1.1.3).

### Determining the contributions of the environmental factors

We used the methodology from Barton et al.[12] to determine the contributions of environmental factors to the alteration of ecological status in the surface ocean induced by climate change. We calculated the probability ($P$) for each ecological status ($n$) under two conditions: i) all ten environmental factors ($E$) using the data from 2023 (present) and ii) all factors using the data from 2023 (present), except for factor $e$, for which we used the data for 2100 (future). We then calculated the relative contribution of factor $e$ ($RC_e$) to the alteration in ecological status induced by climate change at each grid point with a 0.5° × 0.5° spatial resolution using Eq. (3):

$$RC_e = \frac{\sum_{n \in N} \left| P_n^{\text{future for factor } e \text{ only}} - P_n^{\text{present}} \right|}{\sum_{e \in E} \sum_{n \in N} \left| P_n^{\text{future for factor } e \text{ only}} - P_n^{\text{present}} \right|} \tag{3}$$

This equation first calculates the total changes in the probability of ecological status when only one environmental factor (for example, salinity) changes in the future. Then, such changes under different environmental factors were added together to calculate the relative contribution of one environmental factor (such as salinity). Finally, the dominant factor at each grid point was defined as the factor with the highest relative contribution at this point.

### Development of the ES_predictor

For better use of ecological status, we packed our prediction scripts and related files (such as the final classification model and Python environmental files for XGBoost) by using 'pyinstaller' Python package, and developed a software called ES_predictor, which will be licensed under a Creative Commons Attribution 4.0 International license (https://creativecommons.org/licenses/by/4.0/legalcode.en). ES_predictor is an easy-to-use tool developed for Windows users and can be used for determining the ecological status of surface ocean

ecosystems with environmental variables. The software package, demonstrations, and instructional manuals can be obtained via https://doi.org/10.6084/m9.figshare.25627293.

## Statistical analysis and visualization

Significant differences were identified using multiple methods. The *adonis* test (two-sided) with 999 permutations was performed with the 'vegan' R package (v2.5–7) to determine the temporal and spatial variation in both the structure and function of the microbial communities. The Kruskal–Wallis *H* test with pairwise comparisons (Dunn's test) and Cohen's d calculation were performed with the 'scipy' Python package (v1.13.0) to further evaluate the hierarchical clustering performance. The Friedman test with Nemenyi pairwise comparisons was performed with the 'scipy' Python package (v1.13.0) to determine the different performances of the five machine learning regression algorithms under the best hyperparameter combinations. All these statistical tests were two-sided, and the *p* values were adjusted by Bonferroni correction. All the maps were visualized on a world map using the 'Basemap' Python package with the same settings in latitude and longitude. Thus, for concision of the figures, we only defined the latitude and longitude in Fig. 1a, and the others were the same. Plots of the principal component analysis of ecological status were visualized using the 'matplotlib' Python package. Point plots, density plots, line plots, and box plots were visualized using the 'ggplot2' R package. Heatmaps were constructed using TBtools[58] (v2.0.42). All schematic diagrams and elements in Fig. 3f and Supplementary Fig. 12 were created with BioRender.com released under a Creative Commons Attribution-NonCommercial-NoDerivs 4.0 International license (https://creativecommons.org/licenses/by-nc-nd/4.0/legalcode.en). All other plots (e.g., pie plots and histograms) were generated using GraphPad Prism (v8.0.2).

## Reporting summary

Further information on research design is available in the Nature Portfolio Reporting Summary linked to this article.

## Data availability

The sequence data were collected from Bio-GO-SHIP[21], which is publicly available. Information for all the metadata, including the accession codes, is provided in Supplementary Data 1. Supplementary Data containing the critical supplementary information in this study are publicly available online (https://doi.org/10.6084/m9.figshare.25828267). The raw data underlying the figures are provided as Source data that can be obtained from a public repository (https://doi.org/10.6084/m9.figshare.25828276.v2). The sequences in our database of biogeochemical marker genes are available at https://doi.org/10.6084/m9.figshare.25634544.v1. Source data are provided with this paper.

## Code availability

The scripts used in this study are all available online at https://github.com/ZhenyanZhang/Ecological_status[52]. The easy-to-use tool ES_predictor can be downloaded from https://doi.org/10.6084/m9.figshare.25627293.

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

## Acknowledgements

We appreciate the researchers at Bio-GO-SHIP[21] for their work on the sampling of metagenome data from surface oceans and providing them with public resources. All schematic diagrams and elements in Fig. 3f and Supplementary Fig. 12 were created with BioRender.com released under a Creative Commons Attribution-NonCommercial-NoDerivs 4.0 International license (https://creativecommons.org/licenses/by-nc-nd/4.0/legalcode.en). H.F.Q. was financially supported by the National Natural Science Foundation of China (21976161, 22241603-3 and 22376187), the National Key Research and Development Program of China (2022YFD1700401) and the Zhejiang Provincial Natural Science Foundation (LZ23B070001). J.P. was financially supported by the Spanish Government projects (TED2021-132627B-I00 and PID2022-140808NB-I00) funded by MCIN, AEI/10.13039/501100011033 and European NextGenerationEU/PRTR, the Fundación Ramón Areces project (CIVP20A6621) and the Catalan government project (SGR2021-1333). Z.W.F. was financially supported by the National Key Research and Development Program of China (2022YFD1700400 and 2017YFD0200503). Q.Z. was financially supported by the National Natural Science Foundation of China (42307158).

## Author contributions

Zhenyan Zhang and Qi Zhang designed the study with guidance from Haifeng Qian. Zhenyan Zhang wrote the first draft of the manuscript, and Haifeng Qian, Yong-Guan Zhu, Josep Penuelas, Ye Deng and Zhengwei Fu contributed substantially to the revisions. Tingzhang Wang and Xiaoji Fan contributed to the functional annotation of all the metagenome samples. Zhenyan Zhang and Bingfeng Chen performed all the metagenomic analyses. Nuohan Xu and Yitian Yu were responsible for the machine learning model construction and related data analysis. Zhenyan Zhang and Qi Zhang performed the visualization of all data and the artistic design of all figures. Haifeng Qian, Josep Penuelas, Zhengwei Fu and Qi Zhang acquired funding for this project.

## Competing interests

The authors declare no competing interests.
