## [Peer Review File · Nature Communications]

Global biogeography of microbes driving ocean ecological status under climate changeREVIEWER COMMENTS

Reviewer #1 (Remarks to the Author):

This study focusses on a significant region of Earth addressing how global biogeographic patterns of microbes are related to ecological functioning under scenarios of climate change. Previous work has addresses questions regarding the diversity, biogeography, temporality, function and relation to environment of global microbial populations. Here, authors propose an indicator to assess the ecological status of marine microbial communities. Authors worked with a vast dataset of 1433 ocean samples that came from two metagenomic projects (Bio-GO-SHIP and BioGEOTRACES). Authors analyzed this data for functional determinations of biogeochemically relevant pathways, correlating to species composition. This synthesis was compiled as the ecological status per species, accordingly to their functional diversity. Using machine learning techniques, authors included global change scenarios to compute the changes in distribution, abundance, and presence of the microbial communities analyzed. The approach is very interesting, and using machine learning to ring together datasets from metagenomics, climate models, biogeography is clearly a fantastic tool with great potential. This study corroborated the basin-wide differences in biogeography and function of microbial communities that have been previously reported (e.g. Tara Ocean) incorporating finer points such as the relevance of potential elemental-cycling with latitudinal change, which suggests the stability of functional traits and redundancy, which is relevant to understand.

Comments:

Fig S8. Check caption-

Fig S11. Those variables define density and ocean regions- currents- hence, diversity patterns will also be related.

Would it be possible to dig deeper regarding the functional potential following the categories proposed by authors for C, N, P, S cycles? E.g. in Fig. 1, the relative abundance is shown but not the genes or metabolic features associated to each major element.

Are the genetic markers for each element analyzed in flux-models to understand if metabolisms are complete? Or is the dataset organized in presence/absence of genes without correlation to metabolic networks?

The delimitation of oceanic status-provinces accounts for “The structure, diversity, and functionality of a microbial community can vary across the wide range of global environmental conditions which can be determined by the combination of natural processes and anthropogenic activities in the oceans” lines 182-183

The distribution, structure and function also vary temporarily and spatially. The datasets included in this study do not represent this. What implications does this have for the proposed model?

Are the sampling campaigns comparable?

Where samplings done in similar moments?

Lines 245-246. The 245 increase of element-cycling potential under high GHG emission level scenario may 246 result from the increase of element cycling within specific pathways.

Although, Cyanobacteria are limited under current CO₂ppm, hence under GHG emission scenarios, the increase in photosynthetic potential would clearly affect the community structure and function.

Lines 251-259. E.g. The Atlantic basin is changing due to Sargassum blooms- how is this affecting the proposed model? Is this considered?

The methods section needs to be refined since metabolic networks and hence functional classifications cannot be established from presence/absence of gene surveys- there are network flow analysis needed to understand if the pathways are present or not, and their relative abundance. This is fundamental since ecological status considered cycling potential.

Also, details are needed regarding the machine learning assumptions used to infer and model ecological status. Authors mention six variables, but these are not defined, explored or analyzed in this study.

The sampling protocol for the metagenomic datasets also needs further detail since authors state that biases were accounted for but need explain the considerations.

Reviewer #2 (Remarks to the Author):

This paper unifies metagenomes from two global survey studies, and measures several metrics along diversity, composition, and elemental cycling capability as inferred from a custom database of 'element cycling genes' for the C/N/P/S cycles. They then design a set of manual rules to divide these samples into 'ecological status' clusters, and train a random forest classifier to predict the cluster label from environmental data.

The authors convey a passion for the urgency of understanding the effects of climate change on the microbial communities that drive biogeochemical cycling in the ocean, and I commend them for this goal. Unfortunately, they fall short of this objective in their metric design, and in many ways claim results that are not supported by their data.

I draw attention to the particular areas that can be improved below:

Analysis of differences across ocean basins:

-The authors divided their samples into Atlantic, Pacific, and Indian Ocean regions (line 108). I don't see why. What's the motivation? Especially given the lackluster statistics that follow:

--The differences they highlight in Fig 1 do not seem supported by their data. The quartiles in their bar charts all overlap, and the abundance differences are minor, especially considering the small catalog of genes they're considering.

--Fig 1C-E should report t-test p-values, if these differences are in fact significant, to support their conclusions here.

Fig S2 – the authors claim there is a correlation with specific species, but their maximum R^2 value is 0.155 and most are ~ 0.01 . That is not a notable correlation, even if the p value is significant. The data is telling us the opposite of what they claim in the paper (line 83).

Same story with the correlation with geographic distance. R^2 of 0.02 is not a positive correlation. (line 106)

The authors then go on to say that there are latitudinal patterns (paragraph starting line 122), but since the Indian Ocean is all low latitudes this could all be an artifact of the different ocean basins. What do the authors think is driving this, latitude or geographic region? How could they differentiate between the two given the confoundedness of their data?

They also stipulate latitudinal differences are due to temperature but don't show this (line 125). Support or move to discussion.

There are several times where the authors make statements that are better suited to interpretation in the discussion than the results section, e.g. lines 128-129.

-

My main objection to this paper is the formulation of the ecological status metric. This metric is constructed completely manually, and is very susceptible to the confounding factors discussed above.

--There are so many clustering approaches you could use to be quantitative here, then you could see how to interpret clusters based off the individual features.

The authors make claims about latitudinal vs temp/salinity differences in their interpretation for climate change, after pointing out latitudinal differences and region differences, and make no effort to disentangle these potentially confounding factors. This is really necessary if you want to make claims about climate change.

I like the way the authors interpret Figure 3, where they break out individual contributors in their ecological status clusters to see how they are projected to change. However, this makes me question the premise of the paper. You could just build separate models to predict each individual feature, and compare predictions of each model under different climate scenarios. This would also enable you to use quantitative values rather than arbitrary categories that have been constructed from a continuous variable, as has been done here.

The authors offer no suggestion on how to interpret the meaning of their ecological status metric. What does it mean to have high diversity and low cycling potential and be dominated by cyanobacteria? Does this have any impact on the function of these ecosystems? I think this is impossible to deduce as the metric is constructed, and makes me question the utility of such a metric.

--Related to this are questions about the meaning of the cycling potential genes and how they should be interpreted. A lot of N acquisition genes and a lot of N remineralization genes would have very different effects on N cycling and function of ecosystems, but would be weighted the same by this metric, I believe.

I again commend the authors for an ambitious effort, despite my serious concerns about the validity of their metric and interpretation of their approach. I do not think this paper is ready for publication, but I hope that by revisiting their approach to clustering, metric design, and data interpretation they can achieve their research objectives in a future paper.

REVIEWER COMMENTS

Reviewer #1 (Remarks to the Author):

This study focusses on a significant region of Earth, addressing how global biogeographic patterns of microbes are related to ecological functioning under scenarios of climate change. Previous work has addresses questions regarding the diversity, biogeography, temporality, function and relation to environment of global microbial populations. Here, authors propose an indicator to assess the ecological status of marine microbial communities. Authors worked with a vast dataset of 1433 ocean samples that came from two metagenomic projects (Bio-GO-SHIP and BioGEOTRACES). Authors analyzed this data for functional determinations of biogeochemically relevant pathways, correlating to species composition. This synthesis was compiled as the ecological status per species, accordingly to their functional diversity. Using machine learning techniques, authors included global change scenarios to compute the changes in distribution, abundance, and presence of the microbial communities analyzed. The approach is very interesting, and using machine learning to ring together datasets from metagenomics, climate models, biogeography is clearly a fantastic tool with great potential. This study corroborated the basin-wide differences in biogeography and function of microbial communities that have been previously reported (e.g. Tara Ocean) incorporating finer points, such as the relevance of potential elemental cycling with latitudinal change, which suggests the stability of functional traits and redundancy, which is relevant to understand.

Response: Thank you for your positive evaluation of the interest, relevance and utility of our study and your insightful comments, which have greatly improved our work. According to your detailed comments, we have made substantial revisions to our work as follows:

- 1) We investigated the functional potential of metabolic networks, including the enzymes and reactions related to each core pathway involved in photosynthesis, carbon fixation, nitrogen metabolism, and sulfur metabolism. To this end, we revised our database to include genes that can unambiguously be used as marker genes for specific pathways (Supplementary Data 2). A previous study demonstrated that these marker genes can be expressed by microbes and contribute to biogeochemical processes in the ocean¹.
- 2) We analysed all the environmental variables used in machine learning. We obtained data on ten environmental factors, including temperature, salinity, partial pressure of carbon dioxide (pCO₂), mixed layer depth, and concentrations of dissolved oxygen and nutrients (nitrate, phosphate, silicate, carbonate, and iron). These factors have also been used in previous studies for evaluating the response of microbial communities to environmental change¹⁻³. In the revised manuscript, we analysed their variations in our dataset (Supplementary Fig. 7), links to microbial profiles (Figs. 1d, e) and changes under future climate change scenarios (Supplementary Figs. 18 and 19).
- 3) We revised our dataset of metagenomic samples and considered only samples from Bio-GO-SHIP⁴, which is an international multidisciplinary project. We also provided the sampling protocol and DNA extraction protocol for the metagenomic dataset and explained how they can minimize sample biases and make the sampling campaigns comparable ('Methods' section; lines 400-420).
- 4) We provided more details in the 'Results and Discussion', such as the impacts of CO₂ on Cyanobacteria and photosynthetic carbon fixation (lines 259-262).

Despite the above revisions, we still obtained similar results to those in the first version: the ecological status of approximately 32.44% of the surface ocean may

undergo changes from the present to the end of this century, assuming no policy interventions, and this proportion could decrease significantly with effective control of greenhouse gas emissions (Fig. 3a, b).

Above all, substantial revisions according to your comment greatly refined our work and made our story much more solid and clearer. Thank you again for your valuable comments.

Comments:

Fig S8. Check caption-

Response: In the revised manuscript, we checked the information of all the figures (e.g., legends and captions) and ensured that they clearly show the results of our work. Thank you for making us notice this.

Fig S11. Those variables define density and ocean regions- currents- hence, diversity patterns will also be related.

Response: We fully agree with your comment. The environmental variables define the density, ocean regions, and currents, as you mentioned. However, the basic logic for our work is as follows (lines 130-136): 1) Environmental variables change on broad spatial (e.g., different ocean regions) and temporal scales, and this progress can undoubtedly be influenced by the combination of natural processes and anthropogenic activities in the ocean⁵. Notably, natural processes (e.g., density and currents) can also be influenced by anthropogenic activities. 2) Changes in environmental conditions can subsequently alter the ecological status (defined by multiple microbial profiles) of the ocean⁶⁻⁸. In this work, we focused on the latter and directly obtained environmental data from GFDL-ESM4, which has already considered the influence of natural processes

and anthropogenic activities on environmental variables under different scenarios. Using these ten environmental variables, we analysed the distribution patterns of the diversity, structure and biogeochemical potential of ocean microbial communities.

Would it be possible to dig deeper regarding the functional potential following the categories proposed by authors for C, N, P, S cycles? E.g. in Fig. 1, the relative abundance is shown, but not the genes or metabolic features associated to each major element.

Response: Thank you for pointing this out. We dug deeper into the functional potential of detailed pathways rather than categories in all analyses of the revised manuscript, including their spatial and temporal variations, definitions of ecological status and future predictions.

Are the genetic markers for each element analyzed in flux-models to understand if metabolisms are complete? Or is the dataset organized in presence/absence of genes without correlation to metabolic networks?

Response: Thank you for raising this question, which is fundamental for our analysis of microbial profiles and the ecological status of the ocean. In the revised manuscript, we curated a set of biogeochemical marker genes in the core pathways involved in photosynthesis, carbon fixation, nitrogen metabolism, and sulfur metabolism. We also provided detailed information on the enzymes and reactions encoded by these genes in metabolic networks (Supplementary Data 2; Fig. 3f).

The delimitation of oceanic status-provinces accounts for “The structure, diversity, and functionality of a microbial community can vary across the wide range of global

environmental conditions which can be determined by the combination of natural processes and anthropogenic activities in the oceans” lines 182-183

The distribution, structure and function also vary temporarily and spatially. The datasets included in this study do not represent this. What implications does this have for the proposed model?

Response: We agree with your comments that the distributions of structure and function vary temporarily and spatially. We added the related analysis to the revised manuscript. The results showed that the richness, Shannon index and abundance of the dominant phyla and biogeochemical marker genes clearly varied among the different samples across temporal and spatial scales (Fig. 1b and Supplementary Figs. 1-6), which resulted from the variation in environmental conditions (Fig. 1d, e and Supplementary Fig. 7), confirming the tight links between environmental conditions and microbial communities. Based on these results, we constructed machine learning models that considered environmental factors as independent variables and microbial profiles and ecological status as dependent variables. In other words, we presumed that changing environmental conditions directly determine the microbial communities rather than the time and site of the samples. This premise ensures that we can predict microbial profiles and ecological status by using changing environmental data.

Are the sampling campaigns comparable?

Response: Yes, they are. We revised our dataset of metagenomic samples and considered only samples from Bio-GO-SHIP⁴, which is an international multidisciplinary project. We also provided the detailed sampling protocol and DNA extraction protocol for the metagenomic dataset and explained how they can minimize sample biases and make the sampling campaigns comparable (‘Methods’ section; lines

400-410).

Where samplings done in similar moments?

Response: In the revised manuscript, we collected metagenomic samples from 8 cruises in Bio-GO-SHIP that travelled from 2011 to 2020 (Fig. 1a and Supplementary Data 1). Therefore, the samplings were not performed at similar times. Based on the tight links between environmental conditions and microbial communities, we constructed machine learning models that considered environmental factors as independent variables and microbial profiles and ecological status as dependent variables. In other words, we presumed that changing environmental conditions directly determine the microbial communities rather than the time and site of the samples. We obtained the environmental factors for each sample according to their detailed sampling time and site to reduce the uncertainty caused by data collection.

Lines 245-246. The 245 increase of element-cycling potential under high GHG emission level scenario may 246 result from the increase of element cycling within specific pathways.

Although, Cyanobacteria are limited under current CO₂ppm, hence under GHG emission scenarios, the increase in photosynthetic potential would clearly affect the community structure and function.

Response: Thank you for pointing this out. We added pCO₂ as one of the key environmental variables for altering ecological status and detailed microbial indices. Changes in environmental conditions (including dramatic increases in pCO₂) under the SSP5-8.5 scenario profoundly increased the abundance of Cyanobacteria in 61.70% of the changed area, mainly in the low-latitude regions (Fig. 3e and Supplementary Fig.

22). We also observed increasing functional potential for photosynthesis and carbon fixation (e.g., photosystem I, photosystem II and the Calvin cycle) (Fig. 3f). These results were consistent with previous studies showing that elevating CO₂ and temperature under high GHG emissions promoted the growth of marine phytoplankton⁹, primary production¹⁰ and carbon fixation¹¹.

Lines 251-259. E.g. The Atlantic basin is changing due to Sargassum blooms- how is this affecting the proposed model? Is this considered?

Response: Thank you for this suggestion. The occurrence of Sargassum blooms in the Atlantic basin is increasing, similar to the occurrence of extreme events in global climatic systems. Both of these factors induce complexity and uncertainty in future environmental conditions, which play a decisive role in our prediction of ecological status. Unfortunately, we cannot obtain available data for Sargassum blooms in the present or future ocean. We mentioned this limitation at the current stage in the discussion (lines 358-361). However, given the urgency of maintaining global ecological integrity and addressing climate change crises¹², we must operate under a degree of uncertainty¹³. This study provides up-to-date information to help scientists and policymakers collaboratively address these global crises.

The methods section needs to be refined since metabolic networks and hence functional classifications cannot be established from presence/absence of gene surveys- there are network flow analysis needed to understand if the pathways are present or not, and their relative abundance. This is fundamental since ecological status considered cycling potential.

Response: Thank you for raising this question, which is fundamental for our analysis

of microbial profiles and the ecological status of the ocean. In the revised manuscript, we curated a set of biogeochemical marker genes in the core pathways involved in photosynthesis, carbon fixation, nitrogen metabolism, and sulfur metabolism. We also provided detailed information on the enzymes and reactions encoded by these genes in metabolic networks (Supplementary Data 2; Fig. 3f).

Also, details are needed regarding the machine learning assumptions used to infer and model ecological status. Authors mention six variables, but these are not defined, explored or analyzed in this study.

Response: We apologize for the lack of these important details. We analysed all the environmental variables used in the revised manuscript. We obtained data on ten environmental factors, namely, temperature, salinity, partial pressure of carbon dioxide (pCO₂), mixed layer depth, and concentrations of dissolved oxygen and nutrients (nitrate, phosphate, silicate, carbonate, and iron). These factors have also been used in previous studies for evaluating the response of microbial communities to environmental change¹⁻³. In the revised manuscript, we analysed their variations in our dataset (Supplementary Fig. 7), links to microbial profiles (Figs. 1d, e) and changes under future climate change scenarios (Supplementary Fig. 18 and 19).

The sampling protocol for the metagenomic datasets also needs further detail since authors state that biases were accounted for but need explain the considerations.

Response: We revised our dataset of metagenomic samples and considered only samples from Bio-GO-SHIP⁴, which is an international multidisciplinary project. We also provided the detailed sampling protocol and DNA extraction protocol for the metagenomic dataset and explained how they can minimize sample biases and make

the sampling campaigns comparable ('Methods' section; lines 400-410).

Reviewer #2 (Remarks to the Author):

This paper unifies metagenomes from two global survey studies, and measures several metrics along diversity, composition and elemental cycling capability as inferred from a custom database of 'element cycling genes' for the C/N/P/S cycles. They then design a set of manual rules to divide these samples into 'ecological status' clusters, and train a random forest classifier to predict the cluster label from environmental data.

The authors convey a passion for the urgency of understanding the effects of climate change on the microbial communities that drive biogeochemical cycling in the ocean, and I commend them for this goal. Unfortunately, they fall short of this objective in their metric design, and in many ways, claim results that are not supported by their data.

I draw attention to the particular areas that can be improved below:

Response: Thank you for your appreciation of our research goal and your valuable comments, which greatly improved our work and made our manuscript more sound and clearer. We have reanalysed all the data and rewritten the manuscript according to your comments as follows:

- 1) We used hyperparameter-tuned hierarchical clustering rather than manual clustering to cluster and define the ecological status in the revised manuscript. The silhouette score, an important internal evaluation index in hierarchical clustering, was used to evaluate clustering performance. Finally, the 'ward' clustering algorithm with Euclidean distance was chosen to generate seven clusters (Supplementary Data 6), which were defined as ecological status in this study. The Kruskal–Wallis H test with pairwise comparisons (Dunn's test) and Cohen's d were performed to further evaluate the clustering performance

(Supplementary Data 7). After that, we analysed the detailed global distribution and microbial profiles for each ecological status.

- 2) We constructed a series of reliable regression models using machine learning for each microbial index to quantitatively predict and map detailed microbial profiles in the global ocean (Supplementary Data 5). Additionally, to predict the overall alterations in ecological status in the future ocean, we constructed a classification model for ecological status (Supplementary Data 9). Based on the tight links between environmental conditions and microbial communities, we constructed these machine learning models by considering environmental factors as independent variables and microbial indices and ecological status as dependent variables. Our models simultaneously considered multiple environmental factors rather than a single stressor and were more reliable for predicting future conditions.
- 3) We carefully checked all the statistical analysis in the revised manuscript and provided the p values or original data. Specifically, we calculated Cohen's d to evaluate the extent of difference in microbial indices between different types of ecological status (Supplementary Data 7).
- 4) We revised the discussion to avoid overstating our results. In addition, we divided the 'Results and Discussions' section into two sections in the revised manuscript. This makes our descriptions more precise and clearer.

Despite the above revisions, we still obtained similar results to those in the previous version: the ecological status of approximately 32.44% of the surface ocean may undergo changes from the present to the end of this century, assuming no policy interventions, and this proportion could decrease significantly with effective control of greenhouse gas emissions (Fig. 3a, b).

Above all, substantial revisions according to your comment greatly refined our work and made our story much more solid and clearer. Thank you again for your valuable comments.

Analysis of differences across ocean basins:

-The authors divided their samples into Atlantic, Pacific, and Indian Ocean regions (line 108). I don't see why. What's the motivation? Especially given the lackluster statistics that follow:

--The differences they highlight in Fig 1 do not seem supported by their data. The quartiles in their bar charts all overlap, and the abundance differences are minor, especially considering the small catalog of genes they're considering.

--Fig 1C-E should report t-test p-values, if these differences are in fact significant, to support their conclusions here.

Fig S2 – the authors claim there is a correlation with specific species, but their maximum R^2 value is 0.155 and most are ~ 0.01 . That is not a notable correlation, even if the p value is significant. The data is telling us the opposite of what they claim in the paper (line 83).

Same story with the correlation with geographic distance. R^2 of 0.02 is not a positive correlation. (line 106)

Response: Thank you for pointing out these mistakes in the analysis of differences across ocean basins. In the revised manuscript, Longhurst Province was used to show the spatial variation in microbial profiles more clearly, as it is a long-standing concept of biogeochemical partitioning in the global ocean¹⁴ and has also been used for spatial distribution patterns of chemical conditions¹⁵, animals¹⁶, protists¹⁷ and so on. In addition, we carefully checked all the statistical analyses in the revised manuscript and

provided the p values or original data. Specifically, we calculated Cohen's d to evaluate the extent of difference in microbial indices between different types of ecological status (Supplementary Data 7).

The authors then go on to say that there are latitudinal patterns (paragraph starting line 122), but since the Indian Ocean is all low latitudes this could all be an artifact of the different ocean basins. What do the authors think is driving this, latitude or geographic region? How could they differentiate between the two given the confoundedness of their data?

They also stipulate latitudinal differences are due to temperature but don't show this (line 125). Support or move to discussion.

Response: We completely agree. Latitudes and ocean regions are two spatial factors for the variation in microbial indices and environmental conditions, but their combination increases the confoundedness of the analysis. Different latitudes have highly distinct patterns of terrestrial systems (e.g., the Southern Hemisphere has more ocean regions than does the Northern Hemisphere; the Indian Ocean all has low latitudes), which influences the analysis of latitudinal patterns of microbial profiles. Thus, we did not consider latitude in our revised manuscript.

There are several times where the authors make statements that are better suited to interpretation in the discussion than the results section, e.g. lines 128-129.

Response: Thank you for your valuable suggestions. In the revised manuscript, we divided the 'Results and Discussions' section into two sections. However, potential reasons for some descriptions of our analysis and results need to be explained immediately to validate our analysis and results. Thus, we also included the part of

supplementary descriptions of our results in the 'Results' section. In the 'Discussion' section, we focused on the main findings in our work and their ecological significance. In addition, we discuss the limitations of our work in the current stage and propose some feasible suggestions for future studies. Thank you again for making our descriptions more precise and clearer.

My main objection to this paper is the formulation of the ecological status metric. This metric is constructed completely manually, and is very susceptible to the confounding factors discussed above.

--There are so many clustering approaches you could use to be quantitative here, then you could see how to interpret clusters based off the individual features.

Response: Thank you for your valuable suggestion. We used hyperparameter-tuned hierarchical clustering rather than manual clustering to cluster and define the ecological status in the revised manuscript. This reversion makes the ecological status more meaningful and greatly reduces subjectivity. The silhouette score, an important internal evaluation index in hierarchical clustering, was used to evaluate clustering performance. Finally, the 'ward' clustering algorithm with Euclidean distance was chosen to generate seven clusters, which were defined as ecological status in this study. The Kruskal–Wallis H test with pairwise comparisons (Dunn's test) and Cohen's d were performed to further evaluate the clustering performance (Supplementary Data 7). After that, we analysed the detailed global distribution and microbial profiles for each ecological status.

The authors make claims about latitudinal vs temp/salinity differences in their interpretation for climate change, after pointing out latitudinal differences and region

differences, and make no effort to disentangle these potentially confounding factors. This is really necessary if you want to make claims about climate change.

Response: We apologize for the unclear description. The basic logic for our work is as follows (lines 130-136): 1) Environmental variables change on broad spatial (e.g., different ocean regions) and temporal scales, and this progress can undoubtedly be influenced by the combination of natural processes and anthropogenic activities in the ocean⁵. Notably, natural processes (e.g., density and currents) can also be influenced by anthropogenic activities. 2) Changes in environmental conditions can subsequently alter the ecological status (defined by multiple microbial profiles) of the ocean⁶⁻⁸. This logic suggests that environmental conditions (such as temperature/salinity differences) can naturally differ in different ocean regions; however, climate changes caused by intense anthropogenic activities reshape them.

In the revised manuscript, we analysed variations in environmental conditions in our dataset (Supplementary Fig. 7) and their changes under future climate change scenarios (Supplementary Figs. 18 and 19). Our results also confirmed their contributions to the temporal and spatial variations in the richness, Shannon index and abundance of the dominant phyla and biogeochemical marker genes of ocean microbial communities (Fig. 1d, e and Supplementary Fig. 7).

I like the way the authors interpret Figure 3, where they break out individual contributors in their ecological status clusters to see how they are projected to change. However, this makes me question the premise of the paper. You could just build separate models to predict each individual feature and compare predictions of each model under different climate scenarios. This would also enable you to use quantitative values rather than arbitrary categories that have been constructed from a continuous variable, as has

been done here.

Response: Thank you for your insightful suggestion, which made our analysis much more sound. Following your suggestion, we constructed a series of reliable regression models using machine learning for each microbial index to quantitatively predict and map detailed microbial profiles in the global ocean.

A microbial community can be described by multiple ecological dimensions, such as diversity, structure, and functionality, as outlined in our regression models. These characteristics in turn determine the ecological status of communities in various habitats. In terrestrial ecosystems, Guerra *et al.* defined priority areas for soil nature conservation by synthetically considering soil biodiversity and ecosystem services²⁹. Although some previous research has discussed the diversity, structure and functional traits of microbial communities simultaneously^{7,19,20}, we still lack a systematic indicator to describe the ecological status of microbial communities in the global ocean. This situation hampers our ability to understand the comprehensive influence of anthropogenic activities on microbial communities in ocean systems. Thus, we defined the ecological status of an ocean microbial community not only as representing taxonomic and diversity changes, as was done in previous studies^{14,16}, but also considering the biogeochemical potential of the ocean. In addition, our predictions under different climate change scenarios also demonstrated that alterations in ecological status could be effectively summarized and represented by changes in detailed microbial profiles, including poleward shifts in the main taxa, increases in photosynthetic carbon fixation and decreases in nutrient metabolism (Figs. 3e, f and Supplementary Figs. 21-24). Thus, ecological status can be recognized as a more convenient and comprehensive index for evaluating the influence of anthropogenic activities on microbial communities in ocean systems.

The authors offer no suggestion on how to interpret the meaning of their ecological status metric. What does it mean to have high diversity and low cycling potential and be dominated by cyanobacteria? Does this have any impact on the function of these ecosystems? I think this is impossible to deduce as the metric is constructed, and makes me question the utility of such a metric.

Response: Thank you for pointing this out. We used hyperparameter-tuned hierarchical clustering rather than complete manual clustering to cluster and define the ecological status in the revised manuscript. This revision made the ecological status more meaningful and greatly reduced the subjectivity.

A microbial community can be described by multiple ecological dimensions, such as diversity, structure, and functionality, as outlined in our regression models. These characteristics in turn determine the ecological status of communities in various habitats. In terrestrial ecosystems, Guerra *et al.* defined priority areas for soil nature conservation by considering soil biodiversity and ecosystem services²⁹. Although some previous research has discussed the diversity, structure and functional traits of microbial communities simultaneously^{7,19,20}, we still lack a systematic indicator to describe the ecological status of microbial communities in the global ocean. This situation hampers our ability to understand the comprehensive influence of anthropogenic activities on microbial communities in ocean systems. Thus, we defined the ecological status of an ocean microbial community not only as representing taxonomic and diversity changes, as was done in previous studies^{14,16}, but also considering the biogeochemical potential of the ocean.

Our predictions under different climate change scenarios also demonstrated that alterations in ecological status could be effectively summarized and represented by changes in detailed microbial profiles, including poleward shifts in the main taxa,

increases in photosynthetic carbon fixation and decreases in nutrient metabolism (Figs. 3e, f and Supplementary Figs. 21-24). Thus, ecological status can be recognized as a more convenient and comprehensive index for evaluating the influence of anthropogenic activities on microbial communities in ocean systems.

For better use of ecological status, we provide an easy-to-use tool called ES_predictor (available at <https://doi.org/10.6084/m9.figshare.25627293>), which can be used for determining the ecological status of surface ocean ecosystems with changing environmental variables under diverse anthropogenic pressures depending on the research goals. For example, if researchers have already measured the ten environmental variables in areas under different fishing pressures, they can easily evaluate the comprehensive impacts of fishing on microbial ecology by our tools without metagenomic sequencing, which is costly and requires bioinformatics. Considering the amount of data currently available and in the future, this tool will also be continually refined. (lines 343-352)

--Related to this are questions about the meaning of the cycling potential genes and how they should be interpreted. A lot of N acquisition genes and a lot of N remineralization genes would have very different effects on N cycling and function of ecosystems, but would be weighted the same by this metric, I believe.

Response: We completely agree. We investigated the functional potential of the metabolic networks in the revised manuscript, including the enzymes and reactions related to each core pathway involved in photosynthesis, carbon fixation, nitrogen metabolism, and sulfur metabolism (Supplementary Data 2). This approach was useful and invaluable for overviewing and predicting the genetic potential of microbes in biogeochemical processes of the global ocean and their response to climate change.

However, different genes may affect biogeochemical progress to different extents, as you mentioned, and genetic potential is not completely linked to actual transcriptomic activity, metabolomic composition or element cycling. This limitation of our work can be solved by using multidisciplinary methods, such as transcriptomics, metabolomics and chemometrics. But at present, these approaches still lack international projects with standard sampling protocols. We mentioned this limitation at the current stage in the discussion (lines 373-383).

I again commend the authors for an ambitious effort, despite my serious concerns about the validity of their metric and interpretation of their approach. I do not think this paper is ready for publication, but I hope that by revisiting their approach to clustering, metric design, and data interpretation they can achieve their research objectives in a future paper.

Response: Thank you again for your appreciation of our research goal and your valuable comments, which greatly improved our work and made our story increasingly sound. We have reanalysed all the data and rewritten the manuscript according to your comments. We hope our revisions can convince you of the value of our study and can lead you to recommend its acceptance. Many thanks.

References

- 1 Salazar, G. *et al.* Gene Expression Changes and Community Turnover Differentially Shape the Global Ocean Metatranscriptome. *Cell* **179**, 1068-1083.e1021, doi:<https://doi.org/10.1016/j.cell.2019.10.014> (2019).
- 2 Ban, Z., Hu, X. & Li, J. Tipping points of marine phytoplankton to multiple environmental stressors. *Nature Climate Change* **12**, 1045–1051, doi:10.1038/s41558-022-01489-0 (2022).
- 3 Boyd, P. W., Lennartz, S. T., Glover, D. M. & Doney, S. C. Biological ramifications

- of climate-change-mediated oceanic multi-stressors. *Nature Climate Change* **5**, 71-79, doi:10.1038/nclimate2441 (2015).
- 4 Larkin, A. A. *et al.* High spatial resolution global ocean metagenomes from Bio-GO-SHIP repeat hydrography transects. *Scientific Data* **8**, 107, doi:10.1038/s41597-021-00889-9 (2021).
 - 5 Halpern, B. S. *et al.* Spatial and temporal changes in cumulative human impacts on the world's ocean. *Nature Communications* **6**, 7615, doi:10.1038/ncomms8615 (2015).
 - 6 Cohen, N. R. *et al.* Dinoflagellates alter their carbon and nutrient metabolic strategies across environmental gradients in the central Pacific Ocean. *Nature Microbiology* **6**, 173-186, doi:10.1038/s41564-020-00814-7 (2021).
 - 7 Sommeria-Klein, G. *et al.* Global drivers of eukaryotic plankton biogeography in the sunlit ocean. *Science* **374**, 594-599, doi:10.1126/science.abb3717 (2021).
 - 8 Saunders, J. K. *et al.* Microbial functional diversity across biogeochemical provinces in the central Pacific Ocean. *Proceedings of the National Academy of Sciences* **119**, e2200014119, doi:10.1073/pnas.2200014119 (2022).
 - 9 Riebesell, U., Wolf-Gladrow, D. A. & Smetacek, V. Carbon dioxide limitation of marine phytoplankton growth rates. *Nature* **361**, 249-251, doi:10.1038/361249a0 (1993).
 - 10 Hein, M. & Sand-Jensen, K. CO₂ increases oceanic primary production. *Nature* **388**, 526-527, doi:10.1038/41457 (1997).
 - 11 Riebesell, U. *et al.* Enhanced biological carbon consumption in a high CO₂ ocean. *Nature* **450**, 545-548, doi:10.1038/nature06267 (2007).
 - 12 Cavicchioli, R. *et al.* Scientists' warning to humanity: microorganisms and climate change. *Nature Reviews Microbiology* **17**, 569-586, doi:10.1038/s41579-019-0222-5 (2019).
 - 13 Schemm, S. *et al.* Learning from weather and climate science to prepare for a future pandemic. *Proceedings of the National Academy of Sciences* **120**, e2209091120, doi:10.1073/pnas.2209091120 (2023).
 - 14 Hörstmann, C. *et al.* Microbial diversity through an oceanographic lens: refining the concept of ocean provinces through trophic-level analysis and productivity-specific length scales. *Environmental Microbiology* **24**, 404-419, doi:https://doi.org/10.1111/1462-2920.15832 (2022).
 - 15 Li, Z., Zhang, Y. G., Torres, M. & Mills, B. J. W. Neogene burial of organic carbon

- in the global ocean. *Nature* **613**, 90-95, doi:10.1038/s41586-022-05413-6 (2023).
- 16 Bird, C. S. *et al.* A global perspective on the trophic geography of sharks. *Nature Ecology & Evolution* **2**, 299-305, doi:10.1038/s41559-017-0432-z (2018).
- 17 Biard, T. *et al.* *In situ* imaging reveals the biomass of giant protists in the global ocean. *Nature* **532**, 504-507, doi:10.1038/nature17652 (2016).

REVIEWERS' COMMENTS

Reviewer #1 (Remarks to the Author):

In this study, Qian et al., focus on a significant region of Earth addressing how global biogeographic patterns of microbes are related to ecological functioning under scenarios of climate change. I want to congratulate authors for taking in to account all concerns raised during the first round of review. Their article is now more solid and clear. This is fundamental since authors make predictions of biogeochemical function shift under different scenarios of Global Climate Change, hence, the verification of metabolic pathways and methods was fundamental. The approach is very interesting, and using machine learning to bring together datasets from metagenomics, climate models, biogeography is clearly a fantastic tool with great potential.

Reviewer #2 (Remarks to the Author):

The authors have extensively addressed the comments in my previous review, and I think the revised manuscript is much stronger. I still think the ecological status clusters are hard to interpret in terms of functional effects of changes in distributions due to climate change, but this is a limitation for the field as a whole that they sufficiently address in the discussion.

The only thing I want to point out is there is a sentence fragment in describing the datasets used for training the random forest classifier at lines 538-540, "Principal component analysis and correlation metrics between features were used to ensure the validity of the randomly sampled dataset (Supplementary Figs. 14 and 15), which was then used as a training dataset in machine learning and used as a test dataset." Given the very high accuracy they should doublecheck they used an independent test set, it isn't clear from this description.

Aside from that minor comment, I think this manuscript is much improved and I have no other concerns to raise before publication.

REVIEWER COMMENTS

Reviewer #1 (Remarks to the Author):

In this study, Qian et al., focus on a significant region of Earth addressing how global biogeographic patterns of microbes are related to ecological functioning under scenarios of climate change. I want to congratulate authors for taking in to account all concerns raised during the first round of review. Their article is now more solid and clear. This is fundamental since authors make predictions of biogeochemical function shift under different scenarios of Global Climate Change, hence, the verification of metabolic pathways and methods was fundamental. The approach is very interesting, and using machine learning to bring together datasets from metagenomics, climate models, biogeography is clearly a fantastic tool with great potential.

Response: Thank you for your time and effort in handling our manuscript. Your previous comments have greatly improved our work.

Reviewer #2 (Remarks to the Author):

The authors have extensively addressed the comments in my previous review, and I think the revised manuscript is much stronger. I still think the ecological status clusters are hard to interpret in terms of functional effects of changes in distributions due to climate change, but this is a limitation for the field as a whole that they sufficiently address in the discussion.

Response: Thank you for your time and effort in handling our manuscript. We appreciate your previous comments, which greatly refined our work and made our story

much more solid and clearer. In this study, we propose a new concept, ecological status, for ocean biogeography by integrating diversity, biogeochemical potential, and dominant microbial taxa. Ecological status can be recognized as a more convenient and comprehensive index for evaluating the influence of anthropogenic activities on microbial communities in ocean systems.

The only thing I want to point out is there is a sentence fragment in describing the datasets used for training the random forest classifier at lines 538-540, "Principal component analysis and correlation metrics between features were used to ensure the validity of the randomly sampled dataset (Supplementary Figs. 14 and 15), which was then used as a training dataset in machine learning and used as a test dataset." Given the very high accuracy they should doublecheck they used an independent test set, it isn't clear from this description.

Response: Thank you for pointing this out. We revised this sentence to "Principal component analysis and correlation metrics between features were used to ensure the validity of the randomly sampled dataset (Supplementary Figs. 14 and 15), which was then used as a training dataset in machine learning. The remaining dataset after random sampling was used as an independent test dataset." (lines 538-540)

Aside from that minor comment, I think this manuscript is much improved and I have no other concerns to raise before publication.

Response: Thank you again for your valuable comments.